# Prevalence and predictors of magnesium imbalance among critically ill diarrheal children and their outcome in a developing country

Gazi Md. Salahuddin Mamun[1], Monira Sarmin[2], Aklima Alam[2], Farzana Afroze[2], Lubaba Shahrin[2], Abu Sadat Mohammad Sayeem Bin Shahid[2], Shamsun Nahar Shaima[2], Nadia Sultana[2], Mohammod Jobayer Chisti[2]*, Tahmeed Ahmed[2]

1 Infectious Diseases Division, International Centre for Diarrhoeal Disease Research, Bangladesh (icddr,b), Dhaka, Bangladesh, 2 Nutrition Research Division, International Centre for Diarrhoeal Disease Research, Bangladesh (icddr,b), Dhaka, Bangladesh

* chisti@icddrb.org

## Abstract

Despite having essential roles in maintaining human body physiology, magnesium has gained little attention. We sought to evaluate the prevalence and predictors of magnesium imbalance in diarrheal children admitted to an intensive care unit. This retrospective data analysis was conducted among children admitted between January 2019 and December 2019. Eligible children were categorized by serum magnesium levels that were extracted from the hospital database. Among 557 participants, 29 (5.2%) had hypomagnesemia, 344 (61.8%) had normomagnesemia and 184 (33.0%) had hypermagnesemia. By multivariable multinomial logistic regression, we have identified older children (adjusted multinomial odds ratio, mOR 1.01, 95% CI: 1.004–1.018, p = 0.002) as a predictor of hypomagnesemia. Conversely, younger children (adjusted mOR 0.99, 95% CI: 0.982–0.998, p = 0.02), shorter duration of fever (adjusted mOR 0.92, 95% CI: 0.857–0.996, p = 0.04), convulsion (adjusted mOR 1.55, 95% CI: 1.005–2.380, p = 0.047), dehydration (adjusted mOR 3.27, 95% CI: 2.100–5.087, p<0.001), pneumonia (adjusted mOR 2.65, 95% CI: 1.660–4.240, p<0.001) and acute kidney injury (adjusted mOR 2.70, 95% CI: 1.735–4.200, p<0.001) as the independent predictors of hypermagnesemia. The mortality was higher among children with hypermagnesemia (adjusted mOR 2.31, 95% CI: 1.26–4.25, p = 0.007). Prompt identification and management of the magnesium imbalance among critically ill diarrheal children might have survival benefits, especially in resource-limited settings.

## Introduction

Magnesium is the second most abundant intracellular cation and the fourth most abundant cation overall in the human body [1]. It is very important for human health as ionized magnesium is involved in the interaction of more than 300 enzyme reactions [2]. It is essential for

**Data Availability Statement:** On the basis of recommendation of the Institutional Review Board, the Research Administration of icddr,b has

imposed a restriction on disclosing any personal information of hospitalized patients as this data contain sensitive patient information. However, data generated from icddr,b's electronic patient database can be provided to interested researchers for secondary data analyses upon approval of a Data Licensing Application & Agreement by the icddr,b Data Centre Committee. The data request may be sent to Ms. Shiblee Sayeed (shiblee_s@icddrb.org), Head, Research Administration.

**Funding:** The authors received no specific funding for this work.

**Competing interests:** The authors have declared that no competing interests exist.

various functions like electrolyte homeostasis, stabilization of cell membrane, cell division, maintaining neuromuscular excitability, generation of action potentials, and cardiac function [3, 4]. Magnesium also helps immune systems in fighting against infection through the pathway of inflammatory response and production of nitric oxide [5]. Despite these important functions, magnesium homeostasis is still not always considered clinically an important risk factor against many critical bacterial infections. Thus, magnesium has been considered as "the forgotten electrolyte" [2].

Magnesium was well studied in adults. A few studies reported both hypomagnesemia and hypermagnesemia were common among critically ill children, especially with acute kidney injury, sepsis, and associated with poor outcomes [6, 7]. Diarrhea is an important public health concern, especially among lower- and middle-income countries (LMICs). Acute watery diarrhea leading to dehydration and/or acute kidney injury is an important risk factor for fluid and electrolyte imbalance [8] and is presumed to have an impact on magnesium homeostasis. However, very little is known about the prevalence and role of magnesium homeostasis in diarrheal children. So, our objective was to determine the prevalence, associated factors, and outcome of magnesium imbalance among critically ill children having diarrhea and admitted to the intensive care unit of the world's largest diarrheal disease hospital.

## Materials and methods

### Study site and population

This study was operated in the Dhaka Hospital of the International Centre for Diarrhoeal Disease Research, Bangladesh (icddr,b). According to the icddr,b's annual report 2022, this largest diarrheal hospital provides care and treatment for around 200,000 patients a year. Included patients presented with diarrhea in any age or gender, with or without diarrhea-related complications or comorbidities.

Diarrhea is the entry point for admission into the hospital. Since 2009, this hospital became paperless by operating an electronic patient medical record system to keep all clinical and laboratory data. A detailed description of the Dhaka Hospital of icddr,b has been provided elsewhere [9]. The nine-bedded Intensive Care Unit (ICU) is dedicated to aiding medical care for critically ill patients presenting with respiratory distress and/or cyanosis, hypothermia, sepsis, severe sepsis, septic shock, altered mentation, convulsion, severe pneumonia with or without hypoxemia, or respiratory failure. This unit is enriched with necessary facilities for critical care management including pulse oximeter, cardiac monitors, bedside Glucometer, basic life support instruments, cardiac defibrillators, noninvasive ventilation including locally made bubble continuous positive airway pressure (bubble CPAP) oxygen therapy [10], and mechanical ventilators, etc.

### Study design

This is a retrospective chart analysis conducted between 1 January 2019 and 31 December 2019. We included children less than 18 years from the intensive care unit investigated for serum total magnesium levels. Total of 594 children were tested during this study period of one year. Among them, 37 (6.2%) were excluded as they didn't have diarrhea. The remaining 557 (93.8%) eligible children were divided into three categories according to serum magnesium level, and were included in the analysis (Fig 1).

Our laboratory-generated reference value of serum magnesium reported as "Total Magnesium" was 0.65–1.05 mmol/L. Any level below the lower limit was considered as hypomagnesemia and above the higher limit was as hypermagnesemia [11].

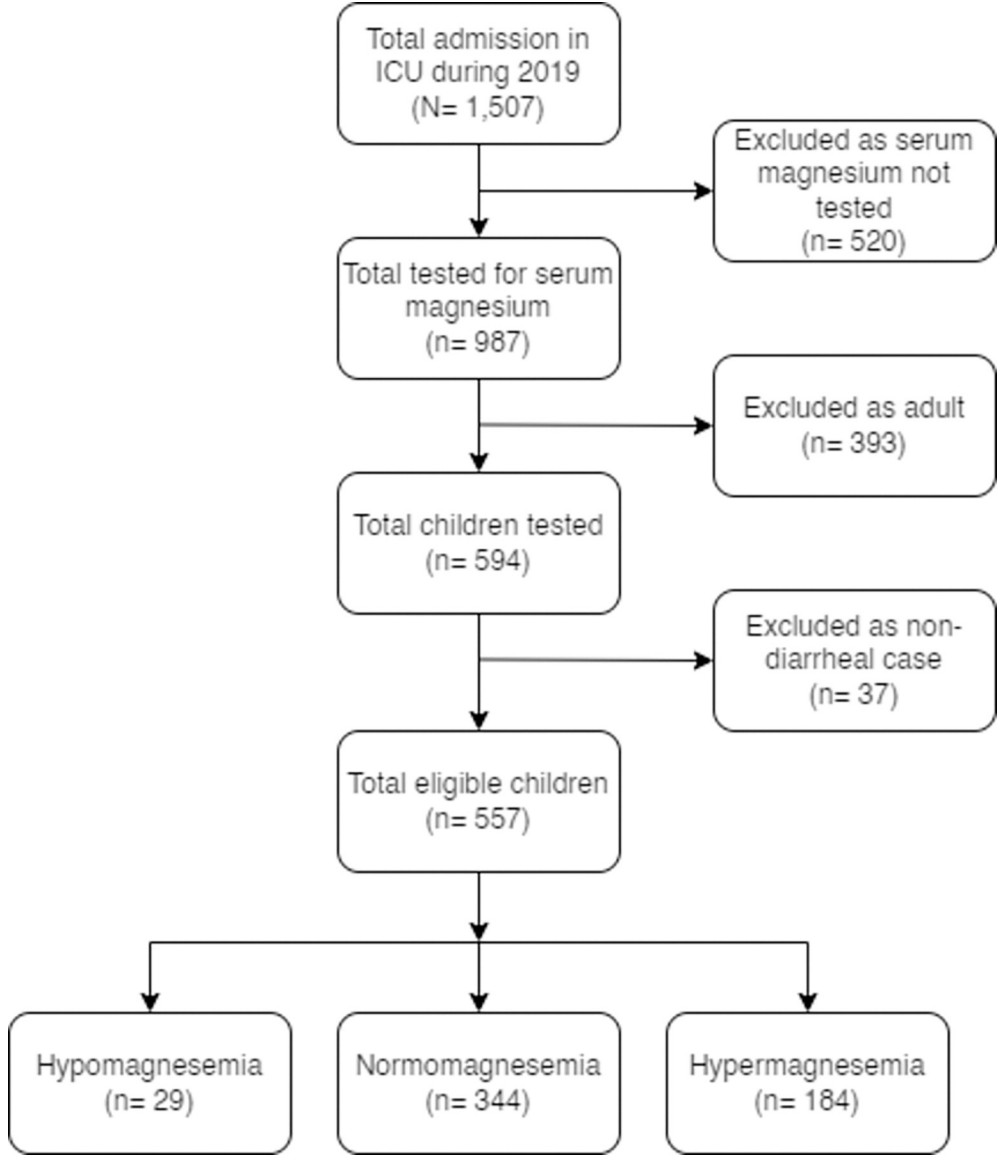

**Fig 1. Study diagram showing the procedure of selection of participants.**

### Patient management

Evidence-based standard treatment protocols were followed during treating dehydrating diarrhea, electrolyte imbalance, and severe pneumonia [9, 12–15]. Children under five presented with severe malnutrition, and diarrhea-related complications were managed according to protocolized management guidelines [12] which is consistent with the WHO [16]. Management of hypoxemia was done using low-cost and locally made bCPAP oxygen therapy [10]. Patients presented with severe dehydration received either cholera saline or normal saline at the Emergency Department and none of these fluids contained magnesium. Serum magnesium was tested after ICU admission of all the participants and none of them received any correction for magnesium imbalance before that.

In addition to these, intravenous fluid resuscitation with isotonic fluid (either Hartmann's solution or Normal saline) was given for patients with severe sepsis. Inotropes and

vasopressors were given in case of septic shock [17]. In case of non-severe patients, only a bolus dose (as mentioned above) was given. Patients were regularly followed up for any adverse reaction such as respiratory distress or flushing of face or restlessness or altered mentation, until discharge.

## Measurements

A semi-structured case report form was developed and finalized for the acquisition of study-relevant data from the electronic database of icddr,b Dhaka Hospital, named Sheba. After preserving the anonymity of the patient, information on sociodemographic status (age, sex), nutritional status, and clinical characteristics on admission such as type of diarrhea, dehydration status, presence of vomiting, fever, pneumonia, respiratory distress, convulsion, sepsis, mental status, presence of hypoxemia were collected. Laboratory test results on admission such as serum sodium, potassium, chloride, bicarbonate, total calcium, total magnesium, and creatinine; treatment history including the use of antibiotics during hospital stay; and variables related to outcome (such as the number of hospital-acquired infections, ventilator support required or not, discharged or referred/left against medical advice or death during hospitalization) were recorded in the paper form.

## Working definitions

Diarrhea was defined as the passage of three or more abnormally loose stools in a day [16]. Dehydration was assessed by the Dhaka method [18]. Fever was considered as the elevation of axillary temperature >38°C. Severe pneumonia and hypoxemia were defined according to the World Health Organization classification [16]. Sepsis, severe sepsis, and septic shock were defined according to surviving sepsis guideline which was adopted for diarrheal patients [17, 19]. The reference value of serum total magnesium was 0.65–1.05 mmol/L. Acute kidney injury was considered if serum creatinine level was more than 1.5 times higher than the normal age & sex-specific serum creatinine level [20].

## Data analysis

All the data were entered into SPSS version 20 (IBM Corp, New York, USA) and analyzed by STATA (version SE 15.0). Clinical, socio-demographic, laboratory and other relevant data were summarized using descriptive statistics. Regarding continuous variables, means with standard deviations were used in case of normally distributed data and medians with interquartile ranges (IQRs) were used in case of skewed data. Categorical data were presented in the frequency table. In this study, a 95% confidence interval and a p-value <0.05 were considered statistically significant values. The strength of association was initially estimated by evaluating the multinomial odds ratio (mOR) by bivariate multinomial logistic regression was used to measure the strength of association and reported as multinomial odds ratio (mOR). Multivariable multinomial logistic regression analysis was done including the variables those were significant to find out the independent predictors of hypomagnesemia and hypermagnesemia compared with normal serum magnesium level.

## Ethical considerations

Data were collected after the extraction of electronic medical records of children hospitalized in the intensive care unit. The information was anonymized and de-identified before analysis so the scope of getting informed written consent was waived. However, waiver of the ethical approval of hospital data disclosure for this study was obtained from the Institutional Review

Board of International Centre for Diarrhoeal Disease Research, Bangladesh (icddr,b). All study procedures and methods were carried out by approved hospital guidelines that were based on the Declaration of Helsinki.

## Results

During the study period, from a total of 1,507 admissions in the intensive care unit, 557 children with diarrhea (37.0%) were tested for serum total magnesium and were included in the analyses. Among them, 29 (5.2%) had hypomagnesemia, 344 (61.8%) had normomagnesemia and 184 (33.0%) had hypermagnesemia (Fig 1). With age, hypermagnesemia had an inverse relationship whereas hypomagnesemia had a linear relationship (Fig 2).

Compared to normomagnesemia, the children with hypomagnesemia were relatively older, more likely to present with sepsis, and had fewer history of convulsion (Table 1). On the other hand, the children with hypermagnesemia were younger, presented with shorter duration of fever, dehydration, pneumonia, hypoxemia, respiratory distress, and raised serum creatinine, and fewer numbers of invasive diarrhea compared with acute watery diarrhea (Table 2). Other variables shown in Tables 1 and 2 were comparable within the groups.

After excluding the inter-related variables to avoid the clinical overfitting having biological plausibility, and adjusting for potential confounders, we did multivariable multinomial logistic regression analysis with the variables significant in any bivariate analysis. We found that older children were more likely to present with hypomagnesemia. Alternately, younger children, shorter duration of fever, convulsion, dehydration, pneumonia and acute kidney injury were significantly and independently associated with hypermagnesemia (Table 3).

Compared to normomagnesemia, serum calcium value was found lower among children with hypomagnesemia (supplementary table 1 in S1 Table) and serum sodium, potassium, and calcium were found higher and acidosis was more prevalent among children with hypermagnesemia (supplementary table 2 in S1 Table).

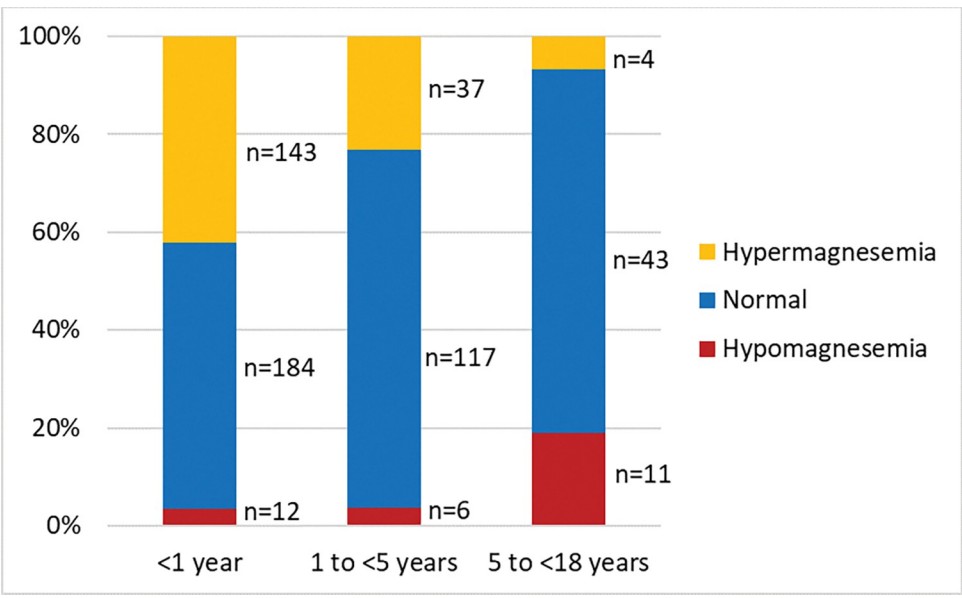

**Fig 2. Distribution of serum magnesium level by age categories.** Here, cut-off levels for hypomagnesemia, normomagnesemia and hypermagnesemia were serum magnesium <0.65 mmol/L, 0.65–1.05 mmol/L and >1.05 mmol/L respectively.

**Table 1. Baseline characteristics of the normomagnesemia and hypomagnesemia children admitted to ICU.**

| Variables | Normomagnesemia (n = 344) | Hypomagnesemia (n = 29) | mOR[1] (95% CI) | P value |
|---|---|---|---|---|
| **Age in months (median, IQR)** | 11 (6, 32) | 31 (6, 168) | 1.01 (1.01–1.02) | <0.001 |
| **Sex (male)** | 201 (58.4) | 19 (65.5) | 1.35 (0.61–2.99) | 0.458 |
| **Type of diarrhea** | | | | |
| Acute watery diarrhea | 227 (66.0) | 19 (65.5) | Reference | |
| Invasive diarrhea | 94 (27.3) | 6 (20.7) | 0.76 (0.30–1.97) | 0.576 |
| Persistent diarrhea | 23 (6.7) | 4 (13.8) | 2.08 (0.65–6.63) | 0.217 |
| **Duration of diarrhea in days (median, IQR)** | 3 (2, 5) | 2 (1, 5) | 1.01 (0.95–1.08) | 0.766 |
| **Vomiting** | 171 (49.7) | 20 (69.0) | 2.25 (0.996–5.08) | 0.051 |
| **Dehydration** | 93 (27.0) | 6 (20.7) | 0.70 (0.28–1.78) | 0.459 |
| **Temperature (˚C) (mean, SD)** | 37.6 ± 1.3 | 37.9 ± 1.3 | 1.21 (0.91–1.62) | 0.198 |
| **Duration of fever in days (median, IQR)** | 2 (1, 4) | 1 (1, 2) | 0.92 (0.79–1.07) | 0.265 |
| **Respiratory distress** | 131 (38.1) | 15 (51.7) | 1.74 (0.81–3.73) | 0.152 |
| **Hypoxemia** | 88 (25.6) | 9 (31.0) | 1.31 (0.57–2.98) | 0.521 |
| **Altered mental status** | 193 (56.1) | 14 (48.3) | 0.73 (0.34–1.56) | 0.417 |
| **Pneumonia** | 163 (47.4) | 13 (44.8) | 0.90 (0.42–1.93) | 0.791 |
| **Convulsion** | 148 (43.0) | 5 (17.2) | 0.28 (0.10–0.74) | 0.011 |
| **Sepsis** | 132 (38.4) | 18 (62.1) | 2.63 (1.20–5.74) | 0.015 |
| **Acute Kidney Injury** | 113 (32.9) | 9 (31.0) | 0.94 (0.41–2.16) | 0.880 |

[1] mOR: multinomial odds ratio

**Table 2. Baseline characteristics of the normomagnesemia and hypermagnesemia children admitted to ICU.**

| Variables | Normomagnesemia (n = 344) | Hypermagnesemia (n = 184) | mOR[1] (95% CI) | P value |
|---|---|---|---|---|
| **Age in months (median, IQR)** | 11 (6, 32) | 7 (4, 11) | 0.98 (0.97–0.99) | <0.001 |
| **Sex (male)** | 201 (58.4) | 117 (63.6) | 1.24 (0.86–1.80) | 0.249 |
| **Type of diarrhea** | | | | |
| Acute watery diarrhea | 227 (66.0) | 145 (78.8) | Reference | |
| Invasive diarrhea | 94 (27.3) | 31 (16.9) | 0.52 (0.33–0.81) | 0.005 |
| Persistent diarrhea | 23 (6.7) | 8 (4.4) | 0.54 (0.24–1.25) | 0.152 |
| **Duration of diarrhea in days (median, IQR)** | 3 (2, 5) | 3 (2, 5) | 0.98 (0.94–1.02) | 0.349 |
| **Vomiting** | 171 (49.7) | 101 (54.9) | 1.23 (0.86–1.76) | 0.257 |
| **Dehydration** | 93 (27.0) | 118 (64.1) | 4.83 (3.29–7.08) | <0.001 |
| **Temperature (˚C) (mean, SD)** | 37.6 ± 1.3 | 37.6 ± 1.2 | 1.03 (0.90–1.19) | 0.674 |
| **Duration of fever in days (median, IQR)** | 2 (1, 4) | 2 (1, 3) | 0.93 (0.87–0.99) | 0.033 |
| **Respiratory distress** | 131 (38.1) | 116 (63.0) | 2.77 (1.92–4.02) | <0.001 |
| **Hypoxemia (SpO2 <90% in room air)** | 88 (25.6) | 74 (70.2) | 1.96 (1.34–2.87) | 0.001 |
| **Altered mental status** | 193 (56.1) | 113 (61.4) | 1.25 (0.86–1.79) | 0.239 |
| **Pneumonia** | 163 (47.4) | 131 (71.2) | 2.74 (1.87–4.03) | <0.001 |
| **Convulsion** | 148 (43.0) | 89 (48.4) | 1.24 (0.87–1.78) | 0.240 |
| **Sepsis** | 132 (38.4) | 69 (37.5) | 0.96 (0.67–1.39) | 0.844 |
| **Acute Kidney Injury** | 113 (32.9) | 125 (67.4) | 4.23 (2.86–6.25) | <0.001 |

[1] mOR: multinomial odds ratio

**Table 3. Association of hypomagnesemia and hypermagnesemia with normomagnesemia by multivariable multinomial logistic regression.**

| Variables | Hypomagnesemia (n = 29) vs Normomagnesemia (n = 344) | | Hypermagnesemia (n = 184) vs Normomagnesemia (n = 344) | |
|---|---|---|---|---|
| | Adjusted mOR[1] (95% CI) | P value | Adjusted mOR (95% CI) | P value |
| **Age (months)** | 1.01 (1.004–1.018) | 0.002 | 0.99 (0.982–0.998) | 0.020 |
| **Duration of fever (day)** | 0.87 (0.726–1.053) | 0.157 | 0.92 (0.857–0.996) | 0.040 |
| **Convulsion** | 0.33 (0.105–1.025) | 0.055 | 1.55 (1.005–2.380) | 0.047 |
| **Dehydration** | 0.76 (0.254–2.272) | 0.623 | 3.27 (2.100–5.087) | <0.001 |
| **Pneumonia** | 1.25 (0.516–3.011) | 0.625 | 2.65 (1.660–4.240) | <0.001 |
| **Sepsis** | 2.20 (0.911–5.322) | 0.080 | 0.72 (0.458–1.125) | 0.148 |
| **Acute Kidney Injury** | 0.78 (0.309–1.961) | 0.595 | 2.70 (1.735–4.200) | <0.001 |

[1] mOR: multinomial odds ratio

Though we have not found any association in disease course and hospital outcome between normomagnesemia and hypomagnesemia (supplementary table 3 in S1 Table), ventilator support was required more for children with hypermagnesemia and their usual discharge was comparatively low compared to children having normal serum magnesium (supplementary table 4 in S1 Table). After adjusting the potential confounders among the outcome variables, these findings remain same (Table 4).

## Discussion

To our knowledge, this is the first study that evaluated the prevalence, predictors, and outcome of magnesium imbalance in critically ill diarrheal children. This retrospective chart analysis identified the higher prevalence of hypermagnesemia than hypomagnesemia and several independent predictors. Being a younger child, having acute onset of fever, convulsion, dehydration, pneumonia, and raised creatinine were significantly and independently associated with hypermagnesemia. Similarly, hypomagnesemia was significantly associated with older children.

Our study involved critically ill diarrheal children of less than 18 years and the prevalence of hypermagnesemia in this population was 33.0% and hypomagnesemia was 5.2%. A study conducted at Dhaka Hospital of icddr,b from 2010 to 2014, found 3.6% hypomagnesemia among 139 under-five critically ill admitted children [21]. Another study conducted at the same site from 2010 to 2013 among 744 admitted diarrheal patients of ≥16 years old, disclosed

**Table 4. Comparison of disease course of the hypomagnesemia and hypermagnesemia with normomagnesemia children during hospital stay.**

| Variables | Hypomagnesemia (n = 29) vs Normomagnesemia (n = 344) | | Hypermagnesemia (n = 184) vs Normomagnesemia (n = 344) | |
|---|---|---|---|---|
| | mOR (95% CI) | P value | mOR (95% CI) | P value |
| **Ventilator support required** | 2.1 (0.69–6.63) | 0.190 | 1.90 (1.04–3.48) | 0.036 |
| **Duration of ICU stay (median, IQR)** | 1.01 (0.92–1.12) | 0.793 | 1.02 (0.97–1.07) | 0.422 |
| **Outcome**- Discharge | Reference | | Reference | |
| LAMA[1] or referred | 1.95 (0.78–4.90) | 0.155 | 2.60 (1.68–4.04) | <0.001 |
| Death | 2.36 (0.74–7.53) | 0.145 | 2.31 (1.26–4.25) | 0.007 |

[1] LAMA: left against medical advice

that hypomagnesemia was 23.5% [22]. Though both of the findings were similar with the age variation in this study, none of the studies have shown the prevalence of hypermagnesemia. Whereas, a study in Pakistan among 179 children of one month to 15 years old, had observed 44% hypomagnesemia on admission in a pediatric intensive care unit [23]. Though another study from China among 974 septic children of 1 month to 18 years has shown that 25.3% had hypomagnesemia and 6.3% had hypermagnesemia, but the median ages were 47 and 18.7 months in these altered magnesium groups respectively [6] and this age variation also correlates with our study findings. Bandsma et al. have also found more cases of hypermagnesemia than hypomagnesemia among the malnourished children at admission in Kenya and Malawi [24]. Another study from Italy has shown that hypomagnesemia among less than 18 years old children was 9.52% whereas it was 59.01% among the elderly age group [25]. Studies have suggested that magnesium absorption is reduced in elderly age group [26], but this is yet to be confirmed among the older children. However, this might be a cause of reduced serum magnesium level among the older children.

Though the duration of fever was nearly similar to normomagnesemia group, still patient with a relatively shorter duration of fever was found an independent predictor of hypermagnesemia in our study. Even so no study was found related to the association with hypermagnesemia among children, but the aggressive illness leading to the more critical condition of the child might be a reason behind this.

However previous studies have found an association of hypomagnesemia with seizures [27, 28], we've not found any independent association between these. Rather, convulsion was found associated with hypermagnesemia. Studies among adult population have found that severe hypermagnesemia might causes drowsiness, hypotonia, areflexia or coma [27]. In our study, convulsion might also happen due to the coexisting of other electrolytes abnormalities from diarrhea. We have found that hypermagnesemia was also associated with hypernatraemia and studies have shown that hypernatraemia can cause convulsion [9]. So, this might be due to the combined electrolytes imbalances as another study among under-twelve years hospitalized Indian children has found no association of magnesium imbalance with seizures [29].

Study related to dehydration among diarrheal children with hypermagnesemia was not found. In our study, we have found this as an independent predictor. This might be due to excessive fluid loss from diarrhea and/or vomiting leading to volume contraction and hypermagnesemia. Similarly, dehydration might lead to more metabolic acidosis among the hypermagnesemia group. This is also reported in other studies among diarrheal children [21, 30].

We have found pneumonia as a predictor of hypermagnesemia in our study. Dabla et al. have shown that one-third of the hypermagnesemia children had pneumonia, though there was no significant association [29]. Some studies among adult pneumonia patients including SARS-CoV-2 have found that hypermagnesemia was more prevalent and ranged between 16%-54% among COVID-19 pneumonia patients [31, 32], whereas among the other hospitalized patients without pneumonia, it ranged between 5.7%-13.5% [27].

Acute kidney injury on admission was another predictor of hypermagnesemia. A recent study among 3,669 children from a pediatric-specific intensive care database has also found that hypermagnesemia was strongly associated with acute kidney injury as this altered magnesium excretion [7]. They have also found an association of hypermagnesemia with increased 28-day mortality [7]. Yue et al. have also shown a higher proportion of serum creatinine level among the children admitted in pediatric intensive care unit having higher magnesium level [28].

Though sepsis was significantly associated with hypomagnesemia in bivariate analysis, after adjusting the potential confounders, it became borderline significant. An important reason behind the association with hypomagnesemia may be due to the attribution of its immune

system effect on sepsis [1]. Though there are several studies among adult sepsis patients with hypomagnesemia, this is limited among children.

Our observation of the independent association of hypocalcemia with hypomagnesemia is understandable. A study by Limaye et al. 2011 in India, stated that 69% of patients had concomitant hypocalcemia and hypomagnesemia [4]. Historically the evidence of the co-existence of hypocalcemia and hypomagnesemia is frequent [3, 29, 33]. On the other hand, hypermagnesemia was significantly associated with higher serum sodium, chloride, and calcium values along with acidosis. Similarly, a study among children has also found that hypomagnesemia and hypermagnesemia were associated with respectively lower and higher values of serum sodium, potassium, and calcium [29].

However we have not found any significant outcome difference between hypomagnesemia patients compared to normomagnesemia, but hypermagnesemia patients required more support of mechanical ventilator and significantly lower patients with hypermagnesemia were cured, survived, and discharged. Similar finding was also shared by some recent studies such as by Wang et al. among critically ill septic children, where they have mentioned that not hypomagnesemia, rather hypermagnesemia was a predictor of inpatient mortality [6]. Also, Dabla et al. have found that a higher proportion of children with hypermagnesemia needed mechanical ventilation and expired where there was no significant difference in hospital stay with altered serum magnesium levels [29]. Both of these findings are similar to us. A previous study by Broner et al. also mentioned hypermagnesemia as a predictor of high mortality in critically ill pediatric patients [34].

## Limitations

This study was conducted retrospectively from the data collected during admission. Also, one-fifth of the patients had been referred to specialized hospitals due to critical conditions such as acute renal failure requiring emergency dialysis, repeated convlusions not controlled by anti-convulsants, complex congenital anomalies, etc. Their survival outcome couldn't be analyzed in this retrospective analysis as icddr,b physicians did not follow the outcome of the referred patients. Doing a prospective study in a controlled environment could be more effective regarding the outcome of the participants.

## Conclusions

The prevalence of hypomagnesemia among older children and hypermagnesemia among younger children were found to be high. Hypermagnesemia needs more attention than hypomagnesemia in children considering the poor outcome. Hypermagnesemia can be predicted by clinical presentations on admission such as younger age, shorter duration of fever, convulsion, dehydration, pneumonia, and acute kidney injury. Thus, the identification of simple clinically associated factors may help to initiate prompt management of hypermagnesemia (serum total magnesium >1.05 mmol/L) that may further help to prevent morbidity and mortality, especially in resource-limited settings.

## Supporting information

**S1 Checklist. STROBE statement—checklist of items that should be included in reports of *case-control studies*.**
(DOC)

**S1 Table. Supplementary tables 1 to 4.**
(DOCX)

## Acknowledgments

We gratefully acknowledge our core donors for their support and commitment to icddr,b's research efforts. Current donors providing unrestricted support include the Governments of Bangladesh and Canada. We would also like to express our sincere thanks to all clinical fellows, nurses, members of the feeding team, and cleaners of the hospital for their invaluable support and contribution to patient care.

## Author Contributions

**Conceptualization:** Gazi Md. Salahuddin Mamun, Monira Sarmin, Mohammod Jobayer Chisti, Tahmeed Ahmed.

**Data curation:** Gazi Md. Salahuddin Mamun, Aklima Alam, Shamsun Nahar Shaima, Nadia Sultana.

**Formal analysis:** Gazi Md. Salahuddin Mamun, Monira Sarmin.

**Methodology:** Gazi Md. Salahuddin Mamun.

**Supervision:** Mohammod Jobayer Chisti, Tahmeed Ahmed.

**Visualization:** Gazi Md. Salahuddin Mamun.

**Writing – original draft:** Gazi Md. Salahuddin Mamun.

**Writing – review & editing:** Gazi Md. Salahuddin Mamun, Monira Sarmin, Aklima Alam, Farzana Afroze, Lubaba Shahrin, Abu Sadat Mohammad Sayeem Bin Shahid, Shamsun Nahar Shaima, Nadia Sultana, Mohammod Jobayer Chisti, Tahmeed Ahmed.

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
