## [Decision Letter · Decision Letter 0]

11 Oct 2023

PONE-D-23-18444Prevalence and predictors of magnesium imbalance among critically ill diarrheal children and their outcome in a developing countryPLOS ONE

Dear Dr. Chisti,

Thank you for submitting your manuscript to PLOS ONE. After careful consideration, we feel that it has merit but does not fully meet PLOS ONE’s publication criteria as it currently stands. Therefore, we invite you to submit a revised version of the manuscript that addresses the points raised during the review process.

We look forward to receiving your revised manuscript.

Kind regards,

Nattachai Srisawat

Academic Editor

PLOS ONE

Journal Requirements:

Reviewers' comments:

Reviewer's Responses to Questions

**Comments to the Author**

1. Is the manuscript technically sound, and do the data support the conclusions?

Reviewer #1: Yes

Reviewer #2: Yes

2. Has the statistical analysis been performed appropriately and rigorously? 

Reviewer #1: Yes

Reviewer #2: Yes

3. Have the authors made all data underlying the findings in their manuscript fully available?

Reviewer #1: Yes

Reviewer #2: Yes

4. Is the manuscript presented in an intelligible fashion and written in standard English?

Reviewer #1: Yes

Reviewer #2: Yes

5. Review Comments to the Author

Reviewer #1: Thank you for allowing me to comment on this interesting topic.

Major concerns

1. This study well answered the prevalence of dysmagnesemia in critically ill children under 18 years of age, it did also showed that mortality was higher among patients with hypermagnesemia, however there are some more questions need to be answered. Firstly, the time point of measuring magnesium level, this was not clearly shown that ICU admission was hospital admission, so hypermagnesemia was totally caused by the patient condition or medical correction before they were transferred to ICU? This also led to the second question, hypermagnesemia could be the parameter showing that some patients were more severe than others so it showed that the hypermagnesemia related to mortality. As we all know that hypermagnesemia could be due to AKI (decreased excretion), and that would be counted as an organ failure.

2. Many factors would effect the magnesium homeostasis, that was not mentioned in this paper. E.g. older children would have lower intestinal absorption that would explain why in this study older children were found with hypomagnesemia? or the resuscitation fluids might effect the Mg level? and these could lead to the differences seen in this study?

3. In mortality associated with hypermagnesemia, I would be more satisfied to see the data on LAMA and Death of these groups, also the mOR showed interesting ratio but may be the numbers of n were too low to conclude in hypomagnesemia group. I think I would be more information to see outcomes of referred patients

Minor concerns

1. Small punctuation error on line 79.

Reviewer #2: Thank you for the opportunity to review the manuscript entitled 'Prevalence and Predictors of Magnesium Imbalance among Critically Ill Diarrheal Children and Their Outcome in a Developing Country.' In this retrospective review, the authors have highlighted the often-forgotten electrolyte, “magnesium”, and its association with diarrhea. Overall, the paper is well-written and emphasizes an intriguing aspect concerning hypermagnesemia at admission and its clinical implications. I have some comments to hopefully help improve your manuscript.

1.It is very surprising to me that hypermagnesemia has a higher prevalence than hypomagnesemia, and the prevalence of hypomagnesemia is quite low. The authors should discuss this interesting result in greater detail. Perhaps factors such as the study population, diagnostic criteria, or other factors influenced these findings. Additionally, it would be beneficial to discuss these results with those from resource-rich countries.

2.Additionally, it would be valuable to understand the mechanism by which diarrhea induces hypermagnesemia.

3.In my opinion, baseline characteristics should be included in the main manuscript and included hypomagnesemia, normomagnesemia, and hypermagnesemia groups.

4.Could you provide more details about the electrolyte imbalance abnormality that led to seizures in the study?

5.Why is metabolic acidosis more severe in the hypermagnesemia group compared to the hypomagnesemia group? How was an explanation?

6.In the conclusion, the authors should include the cut-off point for hypermagnesemia in this study. This would emphasize the need for prompt management and vigilance in this setting.

7.Moreover, in Figure 2, it would provide a clearer picture by including the cutoff points for hypermagnesemia, normomagnesemia, and hypomagnesemia.

6. PLOS authors have the option to publish the peer review history of their article (what does this mean?). If published, this will include your full peer review and any attached files.

Reviewer #1: No

Reviewer #2: No

---

## [Author Response · Author response to Decision Letter 0]

21 Nov 2023

Reviewer #1: Thank you for allowing me to comment on this interesting topic.

Major concerns

1. This study well answered the prevalence of dysmagnesemia in critically ill children under 18 years of age, it did also showed that mortality was higher among patients with hypermagnesemia, however there are some more questions need to be answered. Firstly, the time point of measuring magnesium level, this was not clearly shown that ICU admission was hospital admission, so hypermagnesemia was totally caused by the patient condition or medical correction before they were transferred to ICU? This also led to the second question, hypermagnesemia could be the parameter showing that some patients were more severe than others so it showed that the hypermagnesemia related to mortality. As we all know that hypermagnesemia could be due to AKI (decreased excretion), and that would be counted as an organ failure.

Response: Thank you for your valuable comments and suggestions that will definitely help to improve the quality of this manuscript.

Patients presented with severe dehydration received either cholera saline or normal saline at the Emergency Department and none of these fluids contained magnesium. Serum magnesium was tested after ICU admission of all the participants and none of them received any correction for magnesium imbalance before that. [added in line: 94-98 of track change version] 

Yes, hypermagnesemia might be due to AKI and we’ve explained this under discussion with some other study findings [line: 253-258 of track change version].

2. Many factors would effect the magnesium homeostasis, that was not mentioned in this paper. E.g. older children would have lower intestinal absorption that would explain why in this study older children were found with hypomagnesemia? or the resuscitation fluids might effect the Mg level? and these could lead to the differences seen in this study?

Response: Thank you for your kind query. We have not found any specific article addressing reduced magnesium among older children. But older children might have reduced magnesium absorption as this is reduced by increasing age and this has been added (line: 222-226 of track change version). Information related to resuscitation fluid is also added (line: 94-98 of track change version) and explained under the previous response.

3. In mortality associated with hypermagnesemia, I would be more satisfied to see the data on LAMA and Death of these groups, also the mOR showed interesting ratio but may be the numbers of n were too low to conclude in hypomagnesemia group. I think I would be more information to see outcomes of referred patients

Response: Thank you for your kind suggestion. Yes, we would also be happier to share the outcomes of referred patients. But as this is a retrospective analysis and icddr,b Dhaka Hospital physicians didn’t follow the outcome of the referred patients, so we couldn’t collect the data and have mentioned this under our limitations (line: 287-289 of track change version).

Minor concerns

1. Small punctuation error on line 79.

Response: Thank you. It has been corrected now (line: 82 of track change version).

Reviewer #2: Thank you for the opportunity to review the manuscript entitled 'Prevalence and Predictors of Magnesium Imbalance among Critically Ill Diarrheal Children and Their Outcome in a Developing Country.' In this retrospective review, the authors have highlighted the often-forgotten electrolyte, “magnesium”, and its association with diarrhea. Overall, the paper is well-written and emphasizes an intriguing aspect concerning hypermagnesemia at admission and its clinical implications. I have some comments to hopefully help improve your manuscript.

Response: Thank you for kindly reviewing our manuscript and for your valuable comments. These will definitely improve the quality of our manuscript. We have addressed your comments as below:

1.It is very surprising to me that hypermagnesemia has a higher prevalence than hypomagnesemia, and the prevalence of hypomagnesemia is quite low. The authors should discuss this interesting result in greater detail. Perhaps factors such as the study population, diagnostic criteria, or other factors influenced these findings. Additionally, it would be beneficial to discuss these results with those from resource-rich countries.

Response: Thank you for your kind suggestion. The scenario here regarding the altered serum magnesium level might be due to that this study has been conducted among children. Hypomagnesemia is more common among the older population and this age variation is explained under discussion with some similar study findings including resource-rich countries (line: 208-226 of track change version).

2.Additionally, it would be valuable to understand the mechanism by which diarrhea induces hypermagnesemia.

Response: Thank you for your valuable comment. Diarrhea was present among all the participants as this study was conducted in a diarrhea hospital. So, from this study, it is not possible to compare the effect of diarrhea on the alteration of serum magnesium. As well, we do not know their baseline serum magnesium level before the diarrheal episode. However, as there were varying degrees of dehydration and acute kidney injury those were likely due to diarrhea, so we’ve mentioned their association with these factors (line: 242-244 and 253-258 of track change version).

3.In my opinion, baseline characteristics should be included in the main manuscript and included hypomagnesemia, normomagnesemia, and hypermagnesemia groups.

Response: Thank you for your opinion. Baseline characteristics tables were included in the main manuscript (Table 1 and Table 2).

4.Could you provide more details about the electrolyte imbalance abnormality that led to seizures in the study?

Response: Thank you for your query. We’ve elaborated and added other study findings regarding the association between other electrolyte imbalances that may also cause seizures (line: 237-240 of track change version).

5.Why is metabolic acidosis more severe in the hypermagnesemia group compared to the hypomagnesemia group? How was an explanation?

Response: Dehydration might lead to more metabolic acidosis by causing volume contraction among the hypermagnesemia group. Now it is explained under discussion (line: 244-246 of track change version).

6.In the conclusion, the authors should include the cut-off point for hypermagnesemia in this study. This would emphasize the need for prompt management and vigilance in this setting.

Response: Thank you for your kind suggestion. We’ve now added the cut-off point for hypermagnesemia under the conclusion (line: 298 of track change version).

7.Moreover, in Figure 2, it would provide a clearer picture by including the cutoff points for hypermagnesemia, normomagnesemia, and hypomagnesemia.

Response: Thank you for your kind suggestion. We’ve now added the cut-off points for hypermagnesemia, normomagnesemia, and hypomagnesemia in the legend of Figure 2 (line: 156-159 of track change version).

---

## [Editor Report · Decision Letter 1]

29 Nov 2023

Prevalence and predictors of magnesium imbalance among critically ill diarrheal children and their outcome in a developing country

PONE-D-23-18444R1

Dear Dr. Chisti,

We’re pleased to inform you that your manuscript has been judged scientifically suitable for publication and will be formally accepted for publication once it meets all outstanding technical requirements.

Kind regards,

Nattachai Srisawat

Academic Editor

PLOS ONE

Additional Editor Comments (optional):

The authors has adequately respond to all queries.
---

## [Editor Report · Acceptance letter]

5 Dec 2023

PONE-D-23-18444R1 

Prevalence and predictors of magnesium imbalance among critically ill diarrheal children and their outcome in a developing country 

Dear Dr. Chisti:

I'm pleased to inform you that your manuscript has been deemed suitable for publication in PLOS ONE. Congratulations! Your manuscript is now with our production department. 

Kind regards, 

on behalf of

Dr. Nattachai Srisawat 

Academic Editor

PLOS ONE